# NUGGET2D: DYNAMIC CONTEXTUAL COMPRESSION FOR SCALING DECODER-ONLY LANGUAGE MODELS

## ABSTRACT

Standard Transformer-based language models (LMs) scale poorly to long contexts. We propose a solution based on dynamic contextual compression, which extends the NUGGET approach of Qin & Van Durme (2023) from BERT-like frameworks to decoder-only LMs. Our method models history as compressed "nuggets" which are trained to allow for reconstruction, and it can be initialized with off-the-shelf models such as LLAMA. We demonstrate through experiments in language modeling, question answering, and summarization that NUGGET2D retains capabilities in these tasks, while drastically reducing the overhead during decoding in terms of time and space. For example, in the autoencoding task, NUGGET2D can shrink context at a 20x compression ratio with a BLEU score of 98% for reconstruction, achieving nearly lossless encoding.

## 1 INTRODUCTION

Standard Transformer-based language models Vaswani et al. (2017) suffer from quadratic computational complexity w.r.t. sequence length, making it challenging to scale to long sequences. Proposed solutions (Tay et al., 2022) include sparsifying attention patterns (Beltagy et al., 2020; Ding et al., 2023) or approximating the attention computation with kernel methods (Choromanski et al., 2021). However, not all these approaches are proven effective for NLP tasks (Qin et al., 2023), and very few of them are applied to large language models (LLMs), such as LLaMA (Touvron et al., 2023a).

We propose a solution called 2-Dimensional **Neu**ral **Agg**lomerative **E**mbedding for **T**ext, or NUGGET2D for short, inspired by the approach of Qin & Van Durme (2023) for BERT-like (Devlin et al., 2019) transformer encoders. Instead of attending to all previous tokens, NUGGET2D allows a decoder-only transformer model to attend to a selected subset of tokens, called "nuggets," thus greatly reducing the computing and memory overhead. This selection can be learned in an unsupervised manner with a residual connection built between nuggets and self-attention modules. Moreover, unlike pattern-based efficient attention methods that evenly chunk the text (Rae et al., 2020), the attention behavior learned by NUGGET2D is linguistically meaningful, naturally splitting the text into subsentential units. We illustrate the architecture of NUGGET2D in fig. 1 in comparison to the recently proposed ICAE (Ge et al., 2023), which appends extra memory slots to the input sequence to be used as a compressed representation.

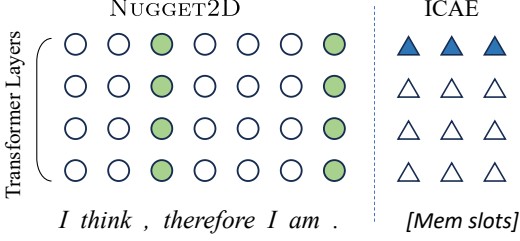

Figure 1: Text compression by ICAE and NUGGET2D . ICAE appends memory slot tokens and uses their last-layer hidden states as the text representation. NUGGET2D adapts the hidden states of a selected subset of input tokens.

NUGGET2D can be trained through autoencoding or next-token prediction objectives, turning it into an efficient context compressor. In experiments on autoencoding, we demonstrate that NUGGET2D can achieve near lossless encoding with a compression ratio as high as 20x, a marked improvement over (Ge et al., 2023, ICAE). After fine-tuning, NUGGET2D is effective in downstream NLP tasks such as question answering (QA) and summarization, where NUGGET2D performs on par with or even better than the original LMs while achieving a compression ratio as high as 10x.

We also show that by re-assigning the parameters in NUGGET2D , we can plug it into LMs and autoregressively aggregate history information into nuggets, delivering an efficient LM. We experimentally demonstrate that NUGGET2D can achieve a perplexity score lower than the original LM with restricted memory, outperforming the baseline model of Rae et al. (2020).

In summary, NUGGET2D adapts Qin & Van Durme (2023) to decoder-only transformers through innovations in its architecture, including a multi-layer memory to compress the history tokens and a residual connection in the self-attention for end-to-end training. In addition, we make it an efficient autoregressive LM by reassigning its parameters. NUGGET2D is shown to be effective in tasks such as autoencoding, language modeling, and applications including QA and summarization.

## 2 APPROACH

In this section, we introduce the architecture of NUGGET2D , including how it adapts NUGGET "1D" into a decoder-only transformer, two of its variants, and the training objective.

### 2.1 BACKGROUND: NUGGET TEXT REPRESENTATION

NUGGET (Qin & Van Durme, 2023) is a text representation method that maps a sequence of $n$ tokens into $k \leq n$ vectors, where a hyperparameter ensures a given compression ratio. E.g., ensuring $k = \frac{n}{10}$. NUGGET first uses a transformer encoder to map $n$ tokens into a sequence of vectors:

$$(\mathbf{x}_1, \mathbf{x}_2, \ldots, \mathbf{x}_n) = \texttt{TransformerEncoder}(w_1, w_2, \ldots, w_n),$$

where $w_i$ and $\mathbf{x}_i \in \mathbb{R}^d$ are the input token and corresponding contextualized vector representation and $d$ is the hidden size of the transformer. A scorer takes $\mathbf{x}_i$ as input and outputs a score $s_i$, where a higher score indicates that the token contains more context information. A select operator picks the top-$k$ vectors as the representation of the input text:

$$(\mathbf{z}_1, \mathbf{z}_2, \ldots, \mathbf{z}_k) = \texttt{TopK}(\mathbf{x}_{1:n}, s_{1:n}, k), \quad s_i = \texttt{Scorer}(\mathbf{x}_i), \tag{1}$$

where $(\mathbf{z}_1, \mathbf{z}_2, \ldots, \mathbf{z}_k)$ are selected from $(\mathbf{x}_1, \mathbf{x}_2, \ldots, \mathbf{x}_n)$ and serve as the resultant representation for the input text. To learn the parameters of Scorer, NUGGET build a residual connection between the encoder and decoder (appendix C). NUGGET was designed for encoder-decoder transformer architectures and was shown to be trainable through autoencoding or machine translation tasks.

### 2.2 NUGGET2D: EFFICIENT TEXT REPRESENTATION IN A DECODER-ONLY LM

Suppose we have a decoder-only transformer LM, such as LLAMA (Touvron et al., 2023a;b), with $L$ self-attention layers. Simplifying, such an LM takes as input the hidden states of past tokens to represent the next token $w_i$ in the $l$-th layer:

$$\mathbf{x}_i^{l+1} = \texttt{Attn}_\phi(\mathbf{x}_i^l, \mathbf{x}_{<i}^l, \mathbf{x}_{<i}^l), \tag{2}$$

where $\mathbf{x}_i^l$ is the vector representation of the $i$-th vector in the $l$-th layer of the transformers, and Attn is a dot-product attention module parameterized by $\phi$ that takes query, key, and value as input. To generate a distribution of the token $w_{i+1}$, an FFN classifier, denoted by LMHead, is applied to the hidden state $\mathbf{x}_i^L$, the last-layer hidden representation of the $i$-th token.

Note that $\mathbf{x}_i^l$ contains not only the information of the current token $w_i$ but also of the past token $w_{<i}$, i.e. they are *contextualized embeddings*. Thus, we hypothesize that attending to a *subset* of past tokens $w_{<i}$, instead of all tokens, is sufficient to support a viable hidden representation $\mathbf{x}_i^l$.

We also hypothesize that some tokens can become *more valuable* than others if only a subset of them can be attended to. Therefore, we propose a scorer that learns to pick out those top-$k$ tokens that should be attended to by future tokens. (see fig. 2) Formally, we have:

$$(\mathbf{x}_{i_1}^l, \mathbf{x}_{i_2}^l, \ldots, \mathbf{x}_{i_k}^l) = \texttt{TopK}(\mathbf{x}_{1:n}^l, s_{1:n}, k), \quad s_i = \texttt{Scorer}_\varphi(\mathbf{x}_i^\lambda), \tag{3}$$

where Scorer is an FFN with parameters $\varphi$, $\mathbf{x}_i^\lambda$ is the $\lambda$-th layer representation of the token $i$ (cf. eq. (1)) and $\lambda$ is a hyperparameter. [1] $(i_1, i_2, \ldots, i_k)$ are the indices of the tokens selected by Scorer.

---

[1] Note $\lambda$ does not depend on $l$, and the selected tokens are the same for all layers.

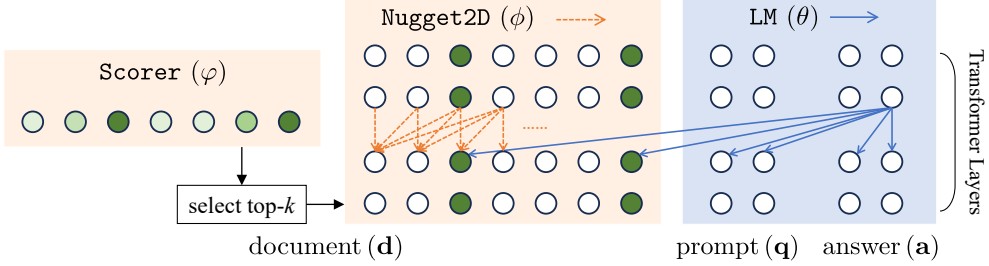

Figure 2: An illustration of how to use NUGGET2D to compress context. From left to right, Scorer select top-$k$ tokens, then Nugget2D encodes the document from the selected tokens, making "nuggets". Finally, LM autoregressively decodes the answer from nuggets and prompt tokens.

We re-write the hidden states of $k$ selected tokens $(\mathbf{x}_{i_1}^l \ldots \mathbf{x}_{i_k}^l)_{l=1}^L$ with $(\mathbf{z}_1^l, \mathbf{z}_2^l, \ldots, \mathbf{z}_k^l)_{l=1}^L$ and use them as compressed representation of the text. [2] As the hidden states for each selected token span over all $L$ layers rather than only the last layer, the approach is named NUGGET2D. In the remainder of this paper, we use $\texttt{Nugget2D}_{\phi,\varphi}$ to encode a sequence of tokens $w_{1:n}$:

$$\mathbf{z}_{1:k} = \texttt{Nugget2D}_{\phi,\varphi}(w_{1:n}), \quad k \leq n, \tag{4}$$

which involves encoding with self-attention ($\phi$) in eq. (2) and token sub-selection ($\varphi$) in eq. (3). The $k$ selected tokens are called *nuggets*, and future tokens may attend to nuggets $\mathbf{z}_{1:k}$ instead of the corresponding tokens $\mathbf{x}_{1:n}$ to access the texts $w_{1:n}$.

Here $k$ depends on the input texts, and we will discuss the selection of $k$ in sections 2.4 and 2.5.

## 2.3 RESIDUAL CONNECTION: ENSURING DIFFERENTIABILITY

The nugget selector relies on the TopK operator to select tokens, which is not differentiable, hindering end-to-end training. Inspired by Qin & Van Durme (2023), we build a residual connection between the nugget scores and the self-attention logits. Suppose that token $\mathbf{x}_i$ attends to nugget $\mathbf{z}_j$, we revise the attention computation as

$$\xi_{i,j}^l = \frac{1}{\sqrt{d}} \left[ \left(\mathbf{W}^Q \mathbf{x}_i^l\right)^\top \left(\mathbf{W}^K \mathbf{z}_j^l\right) + s_j \right], \tag{5}$$

where $\xi_{i,j}^l$ is the weight for the attention from token $\mathbf{x}_i^l$ to nugget $\mathbf{z}_j$ at the $l$-th layer before the normalization of softmax, and $s_j$ is the score of $j$-th nugget: $s_j = \texttt{Scorer}_\varphi(\mathbf{z}_j)$. Therefore, $\texttt{Scorer}_\varphi$ receives gradients via the self-attention module from future tokens at all layers. [3]

To understand how the end-to-end training works, we rewrite the gradients on $s_j$ with the chain rule:

$$\frac{\partial \ell}{\partial s_j} = \sum_{i \in I_j} \sum_{l=1}^L \left( \frac{\partial \xi_{i,j}^l}{\partial s_j} \cdot \frac{\partial \ell}{\partial \xi_{i,j}^l} \right) = \frac{1}{\sqrt{d}} \sum_{i \in I_j} \sum_{l=1}^L \frac{\partial \ell}{\partial \xi_{i,j}^l} \tag{6}$$

where $\ell$ is the loss value and $I_j$ are the indices of tokens that will attend to the nugget $j$. eq. (6) means that the nuggets that are paid more attention by future tokens will by assigned a higher score $s_j$ by $\texttt{Scorer}_\varphi$, averaged across all layers and future tokens. [4]

## 2.4 COMPRESSING CONTEXTS WITH NUGGET2D

One can apply NUGGET2D to compress the context into nuggets. Suppose one asks an LM to decode an answer, denoted by $\mathbf{a}$, from a long context (e.g. a supporting document) and a prompt (e.g. a

---

[2]In the remainder we may omit superscript $l$, using $\mathbf{z}_i$ to denote "2 dimensional" vectors across all layers.

[3]Qin & Van Durme (2023) has a similar mechanism to ensure differentiability but can only be applied to encoder-decoder transformers like BART (Lewis et al., 2020a). The residual connection proposed in this paper is built between any token pairs, and thus can be applied to any transformers.

[4]One can also use straight-through estimator (Bengio et al., 2013) to train Scorer. See appendix B.

query for question answering), denoted by $\mathbf{d}$ and $\mathbf{q}$, they can input the document in the form of nuggets instead of tokens, shown as follows:

$$p(\mathbf{a} \mid \mathbf{d}, \mathbf{q}; \theta, \phi, \varphi) = \prod_i \mathrm{LM}_\theta \left( [\mathrm{Nugget2D}_{\phi,\varphi}(\mathbf{d}); \mathbf{q}; \mathbf{a}_{<i}] \right), \tag{7}$$

where $[\,;\,]$ indicates the concatenation [5] of sequences, LM is a decoder-only LM with the parameters $\theta$ and Nugget2D is the NUGGET2D model with parameters $\phi$ (for transformers) and $\varphi$ (for Scorer). We suppose that $\phi$ and $\theta$ are initialized from the same pretrained LM but they are not tied.

**Choice of $k$**    As the input document is known prior to the compression, the number of nuggets $k$ is set to be proportional to the sequence length: $k = r \cdot n$, where the ratio $r$ is a hyperparameter. Because the LM uses causal masking and Scorer may not naturally select the last token, we require Scorer always selects the last token as a nugget.

**Training objective**    Given a dataset $\mathcal{D}$ with document, prompt, and answer triples, one can train a NUGGET2D model with the next-token prediction objective:

$$(\hat{\theta}, \hat{\phi}, \hat{\varphi}) = \underset{\theta,\phi,\varphi}{\arg\max} \sum_{(\mathbf{d},\mathbf{q},\mathbf{a}) \in \mathcal{D}} \sum_i \log p(\mathbf{a}_i \mid \mathbf{d}, \mathbf{q}, \mathbf{a}_{<i}; \theta, \phi, \varphi). \tag{8}$$

Depending on the task, one may selectively freeze either one or two of the parameter sets among $\theta$, $\phi$, and $\varphi$. Here we discuss 3 situations of training.

*Autoencoding*: NUGGET2D can be trained as an autoencoder, where $\mathbf{d}$ and $\mathbf{a}$ are the same text, while $\mathbf{q} \in \mathbb{R}^d$ is a single-token soft prompt for text reconstruction (Ge et al., 2023; Qin & Eisner, 2021). From the perspective of an autoencoder, $\mathrm{Nugget2D}_{\phi,\varphi}$ is an encoder, the nuggets are the bottleneck, and $\mathrm{LM}_\theta$ is a decoder. All parameters $\theta$, $\phi$, $\varphi$, and $\mathbf{q}$ are trainable.

*Text continuation*: Training $\theta$ risks impacting the language generation performance of the LM (Ramasesh et al., 2021). Therefore, one may freeze $\theta$ and only train $\phi$ and $\varphi$ in eq. (8). A document is split into 3 parts, serving as the $\mathbf{d}$, $\mathbf{q}$, and $\mathbf{a}$. Note the 3 parts may simply be consecutive texts and not necessarily a triple of document, prompt, and answer.

*Fine-tuning for downstream tasks*: One may also make all parameters trainable and fine-tune the LM toward a downstream task. In this scenario, the LM learns to answer the query in the prompt $\mathbf{q}$ by reading the compressed contexts $\mathrm{Nugget2D}(\mathbf{d})$.

## 2.5 AUGOREGRESSIVE NUGGET2D

A common issue with some text compression models (Qin & Van Durme, 2023; Ge et al., 2023) is that they separate the text to be compressed and apply different parameters to the compressor and the decoding module (e.g. $\mathrm{Nugget2D}_{\phi,\varphi}$ and $\mathrm{LM}_\theta$ in section 2.4), which makes it extremely expensive to do autoregressive decoding. Therefore, we introduce a variant of NUGGET2D that restricts the usage of parameter $\phi$ only to the nugget tokens to solve this issue.

The intuition of language modeling with compression is that the distant tokens are less correlated with the next token, and thus can be gradually compressed with NUGGET2D. We illustrate the architecture of autoregressive NUGGET2D in fig. 3. Suppose we have a context of $t$ tokens $w_{1:t}$ and we split them into *distant* tokens $w_{1:\tau}$ and *recent* tokens $w_{\tau+1:t}$, then the distribution of the next token $w_{t+1}$ can be modeled as:

$$(\mathbf{z}_1, \mathbf{z}_2, \ldots, \mathbf{z}_k) = \mathrm{Nugget2D}_{\phi,\varphi}(\mathbf{x}_{1:\tau}), \qquad 1 \le k \le \tau \tag{9}$$

$$\mathbf{x}_t^{l+1} = \mathrm{Attn}_\theta \left( \mathbf{x}_t^l, \left[ \mathbf{z}_{1:k}^l; \mathbf{x}_{\tau+1:t}^l \right], \left[ \mathbf{z}_{1:k}^l; \mathbf{x}_{\tau+1:t}^l \right] \right), \qquad 1 < \tau < t \tag{10}$$

$$p(w_{t+1} \mid w_{1:t}) = \mathrm{LMHead}_\theta \left( \mathbf{x}_t^L \right) \tag{11}$$

where $\tau$ is an index that splits past tokens $w_{1:t}$ into distant and recent tokens, $\mathbf{x}_t^L$ is the last-layer representation of the $t$-th token, and Nugget2D compresses the hidden representation of distant tokens $\mathbf{x}_{1:\tau}$ into $\mathbf{z}_{1:k}$. In eq. (10), the hidden state of the $t$-th token is derived by attending to recent texts as tokens (i.e. $\mathbf{x}_{\tau+1:t}^l$) and attending to distant texts as nuggets (i.e. $\mathbf{z}_{1:k}^l$).

---

[5] The concatenation of nuggets and tokens means concatenating their hidden states.

**Parameter reassignment** In section 2.4, both nugget and non-nugget tokens on the $\texttt{Nugget2D}_{\phi,\varphi}$ side use $\phi$ to parameterize the self attention (eq. (2)), and all tokens on the $\texttt{LM}_\theta$ side use $\theta$. However, an autoregressive NUGGET2D does not have encoder and decoder sides. Instead, the nugget tokens use $\phi$ (in eq. (9)) while the non-nugget tokens use $\theta$ (in eq. (10)) to attend previous tokens. In eq. (9), $\texttt{Nugget2D}_{\phi,\varphi}$ encodes a nugget token $\mathbf{z}_i$ by attending to its previous tokens $\mathbf{x}_{<i'}$ by

$$\mathbf{z}_i^{l+1} = \texttt{Attn}_\phi(\mathbf{z}_i^l, \mathbf{x}_{<i'}^l, \mathbf{x}_{<i'}^l), \tag{12}$$

where $i'$ is the index of the token corresponds to nugget $\mathbf{z}_i$. In contrast, the hidden states of the non-nugget tokens are encoded with $\theta$ as we show in eq. (10).

**Nugget selection** Another function of $\texttt{Nugget2D}_{\phi,\varphi}$ is to select nugget from tokens with $\texttt{Scorer}_\varphi$. Unlike section 2.4, an autoregressive LM does not know the length of sequence. Therefore, we make $k$ dynamic by setting a threshold $\bar{s}$ on the nugget score $s_i$ to select nuggets. $\bar{s}$ is set such that a ratio $r$ of tokens are selected nuggets, averaged over all documents. However, the choice of $\bar{s}$ depends on a trained $\texttt{Scorer}_\varphi$, while training $\texttt{Scorer}_\varphi$ needs a specified $\bar{s}$. To solve this cold-start problem, we reuse the $\texttt{Scorer}_\varphi$ in the autoencoding experiments in section 2.4 and freeze the parameters $\varphi$.

To make the inference time finite for arbitrarily long sequences, we suppose NUGGET2D can read recent $\omega_r$ tokens as tokens and another $\omega_d$ compressed tokens. Texts that are $\omega_r + \omega_d$ tokens away are discarded. In this case, $\tau = t - \omega_r$. We set $\omega_r$ and $\omega_d$ as hyperparameters. Note that although the attention range of each token is restricted to $\omega_r + \omega_d$ tokens, $\mathbf{x}_i^L$ can get access to the information that is $L \cdot (\omega_r + \omega_d)$ tokens away, where $L$ is the number of transformer layers because the information can be retained layer-by-layer (Dai et al., 2019).

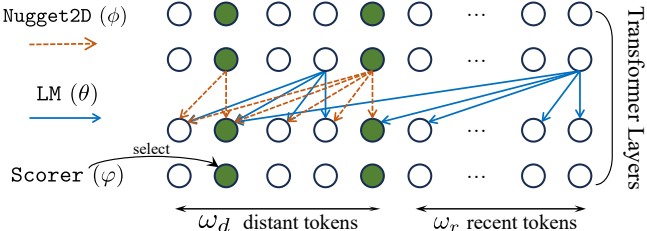

Figure 3: An illustration of the autoregressive NUGGET2D, where $\texttt{Scorer}(\varphi)$ selects nugget tokens, $\texttt{Nugget2D}(\phi)$ aggregates the information of distant tokens into nuggets. When predicting a new token, $\texttt{LM}(\theta)$ has direct access to recent tokens but needs the nuggets to access the distant information.

**Training** We use the next-token prediction as the training objective. However, eqs. (9) and (11) are not easy to parallelize because $\tau$ shifts for every new token. So we approximate eqs. (9) and (11) by sampling sentences with up to $(\omega_d + \omega_r + \omega_p)$ tokens from a corpus $\mathcal{D}$, using the first $\omega_d$ tokens as distant tokens to be compressed by $\texttt{Nugget2D}_{\phi,\varphi}$ and the following $\omega_r$ tokens as recent tokens, and letting the model learn to predict the last $\omega_p$ tokens. Formally, we have:

$$(\hat{\theta}, \hat{\phi}) = \arg\max_{\theta,\phi} \sum_{\mathbf{x} \in \mathcal{D}} \sum_{i=\omega_d+\omega_r}^{\omega_d+\omega_r+\omega_p} \log \texttt{LMHead}_\theta\left(\mathbf{x}_i^L\right) \tag{13}$$

$$\mathbf{x}_i^L = \texttt{LM}_\theta\left(\left[\texttt{Nugget2D}_{\phi,\varphi}(\mathbf{x}_{1:\omega_d}); \mathbf{x}_{\omega_d+1:i}\right]\right). \tag{14}$$

Note that $\varphi$ is not optimized, because we reuse the $\texttt{Scorer}$ in section 2.4 without further fine-tuning.

## 3 OVERALL EXPERIMENT SETUP

We adopt the architecture of LLAMA (Touvron et al., 2023a;b) as our base model. For the autoencoding experiment, we use the checkpoint of $\texttt{LLaMA-7B}$ following ICAE (Ge et al., 2023). We use the checkpoint of $\texttt{LLaMA-2-7B-chat}$ for the downstream NLP tasks and $\texttt{LLaMA-2-7B}$ for the autoregressive language modeling experiments.

We adopt LORA (Hu et al., 2022) with a rank of 32 to fine-tune the parameters of the LM, namely $\theta$ and $\phi$. We used the trick of mixed precision to save GPU memory. Following Qin & Van Durme (2023), we set $\lambda = 3$ and derive $\mathbf{x}_i^\lambda$, the features fed into $\texttt{Scorer}$, from a frozen LM, so the nugget selection is independent of $\theta$ or $\phi$. For the rest of the training details, including training hyperparameters, devices, and parameter count, readers may refer to appendix D.

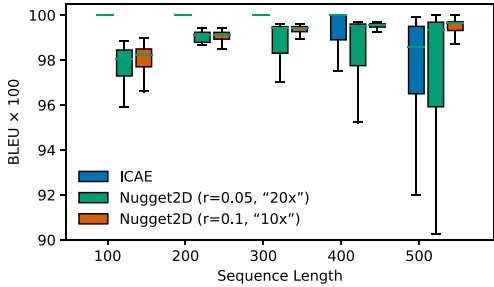
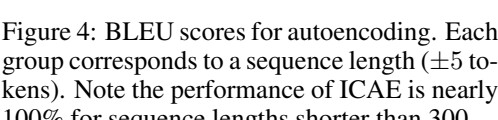
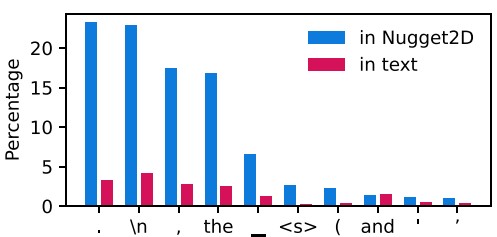

Figure 4: BLEU scores for autoencoding. Each group corresponds to a sequence length ($\pm 5$ tokens). Note the performance of ICAE is nearly 100% for sequence lengths shorter than 300.

Figure 5: Token frequency of nuggets selected by NUGGET2D and the formal texts. These top 10 token types cover 95% of nuggets observed.

# 4 AUTOENCODING EXPERIMENT

## 4.1 TASK DEFINITION, DATASET, AND EXPERIMENT SETUPS

In this section, we use NUGGET2D as a context compressor (section 2.4) and apply it to the autoencoding task. A model is asked to reconstruct the input text from a compressed representation. Following Ge et al. (2023), we fine-tune the `LLaMA-7B` model on the Pile (Gao et al., 2020) dataset. We manually split the corpus into train, dev, and test splits.

As stated in section 2.4, we use NUGGET2D to compress the input text into nuggets, and then use the LM to decode the input sequence. The nugget ratio $r$ is set as 0.05 and 0.1. The soft prompt used for autoencoding is randomly initialized and the length is set as 1.

We use In-Context AutoEncoder (Ge et al., 2023, ICAE) as a baseline model. The key idea of ICAE is to append 128 tokens to the input sequence as "memory slots," (fig. 1) and train the decoder to reconstruct the input from the memories:

$$(\tilde{\mathbf{m}}_1, \tilde{\mathbf{m}}_2, \dots, \tilde{\mathbf{m}}_{128}) = \text{LM}\left([w_{1:n}; m_{1:128}]\right)$$
$$p(w_{i+1} \mid w_{1:i}) = \text{LM}\left([w_{1:i}; \tilde{\mathbf{m}}_{1:128}]\right).$$

We measure using BLEU (Papineni et al., 2002) score on pairs of input and decoded texts. [6]

## 4.2 EXPERIMENT RESULTS

In fig. 4 we see NUGGET2D has comparable performance with the ICAE baseline for short sequences and better performance for long sequences. Moreover, NUGGET2D successfully handles longer inputs: performance improves on longer sequences because the number of nuggets is proportional to the sequence length, unlike ICAE's constant-sized memory. Despite its variable memory, NUGGET2D maintains an advantage over ICAE in computational time and space. First, NUGGET2D *encodes* sequences more efficiently: while ICAE always *appends* 128 tokens, NUGGET2D *reuses* a fraction of the already-encoded tokens. Also, NUGGET2D *uses fewer tokens* than ICAE: even for the longest sequences, NUGGET2D only uses 25 or 50 tokens, while ICAE uses 128 for all sequences. [7] Lastly, NUGGET2D is more efficient than ICAE during *decoding* because it uses fewer tokens and does not need to re-encode them. In short, compared to the baseline, NUGGET2D demonstrates comparable or better performance, successful handling of long sequences, and much more efficient encoding and decoding.

## 4.3 NUGGET TOKENS ARE CLAUSAL TEXT DELIMITERS

In section 2.2, we employ `Scorer` to pick out nuggets, but what are the actual tokens selected? We empirically sampled 128 documents with 50k tokens and run the `Scorer` from the checkpoint

---

[6] Ge et al. (2023) has no associated code, so we report ICAE results per their section 3.3.1.

[7] NUGGET2D uses all layers while ICAE only uses the last layer. However, ICAE needs to encode their memory tokens into hidden states during decoding, while NUGGET2D can save this step.

in section 4 with a nugget ratio of 0.1, and the results are shown in fig. 5. Readers may refer to appendix E for case studies on sampled texts. From fig. 5, we observe similar phenomena as Qin & Van Durme (2023), where the tokens preferred by NUGGET2D are mostly clausal text delimiters, such as punctuation marks, new line symbols, and conjunction words.

## 5 AUTOREGRESSIVE LM EXPERIMENT

### 5.1 EXPERIMENT SETUP

We use the Pile (Gao et al., 2020) and WikiText-103 (Merity et al., 2017) as the corpus for language modeling experiments with autoregressive NUGGET2D . We split the texts into distant and recent segments, where an LM is asked to predict the next token based on recent tokens and compressed distant tokens ( fig. 6). We introduce the method of Compressive Transformers (Rae et al., 2020) as our baseline, which is a method that evenly chunks the text into segments and uses a pooling algorithm to compress the hidden states of each segment into a single vector. We adopt the mean pooling algorithm and make sure they have the same compression ratio as NUGGET2D .

Both models are initialized from the checkpoint `Llama-2-7b` with a compression ratio of 10x. We apply the same training configurations to NUGGET2D and COMPRESSIVE , where we set $\omega_d$, $\omega_r$, and $\omega_p$ as 512, 0, and 128. In addition to those 2 models, we introduce FULL , which is the original LLAMA without any compression. We train all three models on the Pile dataset.

> ... *In the 1890s, armed standoffs were avoided narrowly several times. The Great Northern Railway, under the supervision of president ... (omitted 230 tokens) ... The railway also built Glacier Park Lodge, adjacent to the park on its east side, and the Many Glacier Hotel on the east shore of Swiftcurrent Lake. Louis Hill personally selected the sites for all of these buildings, choosing each for their dramatic scenic backdrops* and views. Another developer, John Lewis, built the Lewis Glacier Hotel on Lake McDonald in 1913–1914. **The Great Northern Railway** bought the hotel in 1930 and it was later ...

Figure 6: An example of a setting of our LM experiment. Here, compressive models access 320 tokens of history (italics) which they must compress to 32 states, along with 32 explicit tokens of most recent history (final portion of red, normal text). FULL gets explicit access only to the entirety of the red text (64 tokens), with no access to longer history. Models need to complete the sequence starting with **The Great Northern Railway**. We would hope both topical and explicit vocabulary information (e.g., the underlined text) will be retained in the compressed history.

### 5.2 EXPERIMENT RESULTS

Table 1: Perplexity on the Pile and WikiText-103, contrasting two 10x compressed solutions against no use of compression. **Cmpr. tokens**: the number of compressed tokens that precede **ctx. tokens**: the uncompressed context immediately before the token to be predicted. This adds up to **total state**, which is directly comparable between systems, using two settings (128 and 64). NUGGET2D trades off explicit context for larger history, with better perplexity results.

| Model | total state | cmpr. tokens | ctx. tokens | ppl. on WikiText *subword* | *word* | ppl. on Pile *subword* |
|---|---|---|---|---|---|---|
| FULL | 128 | 0 | 128 | 6.87 | 11.69 | 5.35 |
| COMPRESSIVE | 128 | 640 | 64 | 7.09 | 12.18 | 4.93 |
| NUGGET2D | 128 | 640 | 64 | **6.58** | **11.06** | **4.49** |
| FULL | 64 | 0 | 64 | 7.95 | 14.08 | 5.80 |
| COMPRESSIVE | 64 | 320 | 32 | 7.64 | 13.39 | 5.65 |
| NUGGET2D | 64 | 320 | 32 | **6.91** | **11.78** | **5.01** |

Perplexity (PPL) is used to evaluate the model performance. Following previous work, we exclude the tokens that are defined as out-of-vocabulary by Merity et al. (2017) from WikiText-103 . Because WikiText is a tokenized corpus, we take production over the probabilities of subwords for each

complete word to measure the word PPL. [8] Because Wikipedia is part of the training corpus of LLaMA, we additionally evaluate the model performance on a held-out subset of the Pile, which is randomly sampled and contains 100k tokens. We report the subword PPL on the Pile data and both subword and word PPLs on WikiText.

The experiment results are shown in table 1. We conduct experiments with 2 context configurations, where an LM has access to up to 64 or 128 past hidden states. For NUGGET2D and COMPRESSIVE, the first 32 or 64 states are compressed representation of the past 320 or 640 tokens. NUGGET2D outperforms both COMPRESSIVE and FULL, showing that with a restricted size of hidden states, NUGGET2D is an effective method to encode history information.

## 6 DOWNSTREAM TASKS: QA AND SUMMARIZATION

### 6.1 EXPERIMENT SETTINGS

**Training objective** We train NUGGET2D with the text continuation objective (section 2.4) using documents sampled from the Pile (Gao et al., 2020). Each document is split into $\mathbf{d}$, $\mathbf{q}$, and $\mathbf{a}$ with up to 512, 16, and 128 tokens respectively (refer to section 2.4). After convergence, we either take it to do downstream tasks in zero-shot settings or further fine-tune the model with $(\mathbf{d}, \mathbf{q}, \mathbf{a})$ triples sampled from the target task. The nugget ratio $r$ is set as 0.1 and 0.2 in this section.

**Baselines** [9]

FULL : Results of the original LM without any compression.
NODOC : LM is used to do the task without any documents, i.e. $(\mathbf{q}, \mathbf{a})$ pairs only.
LMSUMM : Use the LM to summarize the text into fewer tokens with prompts, which are designed to guide the LM to roughly compress the texts into 10% of its length. [10] LM uses the summary instead of documents to do the task. (appendix F)

**Dataset** Datasets used in this section are SQuAD (Rajpurkar et al., 2016) for question answering, and CNN/DailyMail v3.0.0 (See et al., 2017) for summarization (table 2).

Table 2: Dataset statistics. The text lengths are counted by the LLaMA tokenizer.

| Dataset | Split sizes | | | Text length | | |
|---|---|---|---|---|---|---|
| | train | dev | test | doc | prompt | answer |
| SQuAD (Rajpurkar et al., 2016) | 88k | 10.5k | - | 231 | 17.0 | - |
| CNN/DailyMail (See et al., 2017) | 287k | 13.4k | 12k | 878 | - | 68.9 |

### 6.2 QUESTION ANSWERING

In SQuAD a model is asked to extract a phrase from the passage to answer the query. We reformulate this problem as a text-to-text task instead of annotation, using accuracy to evaluate the model performance. As the model tends to generate tokens more than the answer itself or using different forms (e.g. using "two" instead of "2"), we normalize the output to match the answer. Readers may refer to appendix G for the algorithm used to calculate the accuracy.

We consider all models: FULL, LMSUMM, NUGGET2D, and NODOC (table 3).All models are evaluated in a zero-shot manner without fine-tuning. FULL and NUGGET2D easily outperform the NODOC and LMSUMM, and we observe that

Table 3: The accuracy (ACC.) of all 4 models on SQuAD. Cmpr. is the compression of the method.

| Model | Cmpr. | Acc. |
|---|---|---|
| NODOC | $\infty$ | 1.4 |
| LMSUMM | 10x | 30.9 |
| FULL | 1x | **64.5** |
| NUGGET2D | 5x | 59.1 |
| NUGGET2D | 10x | 49.8 |

LMSUMM often omits details that are needed by the question. The performance of NUGGET2D can be improved by lowering its compression ratio, and the performance of NUGGET2D ($r = 0.2$) is close to FULL, confirming a compressed representation can still support LLM reasoning.

---

[8]Note our algorithm for the complete word PPL underestimates the model performance.

[9]We are unable to conduct experiments with ICAE (Ge et al., 2023) because they did not release their code, nor can we run NUGGET2D on their data because this also is unreleased.

## 6.3 SUMMARIZATION

Table 4: Rouge scores ($F_1$ of Rouge-1, Rouge-2, LCS). ZS and FT are zero-shot and fine-tuning.

| Model | Cmpr. | R1 | R2 | RL |
|---|---|---|---|---|
| FULL (ZS) | 1x | 32.5 | 9.7 | 28.2 |
| FULL (FT) | 1x | 37.7 | **15.6** | 35.3 |
| NUGGET2D | 10x | **39.9** | 14.6 | **37.0** |

CNN/DailyMail contains news articles, where a model is required to generate a short summary. As no query is involved, we propose a prompt as a statement of the task requirement (appendix F).

We consider FULL and NUGGET2D ($r = 0.1$). FULL is evaluated in both zero-shot and fine-tuning settings and NUGGET2D is fine-tuned. The results are shown in table 4. We find that NUGGET2D can achieve similar or even better performance than FULL after compression. We speculate that as the context of CNN/DailyMail is long, this may lead the LM to be "lost in the middle" (Liu et al., 2023), whereas the nugget representation generated by NUGGET2D is only 10% of the original length and perhaps less susceptible. This is an interesting avenue for future exploration.

## 7 RELATED WORK AND DISCUSSION

Scaling transformers to long sequences is a popular topic in the NLP community (Tay et al., 2022). Existing work includes sparsify the attention patterns (Beltagy et al., 2020; Zaheer et al., 2020; Khalitov et al., 2023; Ding et al., 2023; Ainslie et al., 2023; Rae et al., 2020), employing low-rank or kernel methods to approximate the attention matrix computation (Choromanski et al., 2021; Katharopoulos et al., 2020), or applying recurrence (Dai et al., 2019; Yang et al., 2019; Bulatov et al., 2022). Another line of work tries to extrapolate the ability of LMs to long contexts, such as using linear bias (Press et al., 2022) or rotary position embeddings (Su et al., 2022). More akin to our work, Bertsch et al. (2023) applied $k$NN search to select a subset of tokens for attention at each layer of an encoder-decoder transformer, effectively extending the attention range of transformers.

In the context of large language models, recent work focuses on compressing the prompt tokens into soft embeddings (Mu et al., 2023; Wingate et al., 2022) or encoding the supporting documents Ge et al. (2023); Chevalier et al. (2023) into fewer vectors. Some recent work tries to train LLMs with longer contexts, such as Li et al. (2023), GLM (Zeng et al., 2022), and Claude 2 (Anthropic, 2023).

Researchers also explored retrieval-based methods that infuse knowledge into LM decoding, some notable work in this field includes FiD (Izacard & Grave, 2021), REALM (Guu et al., 2020), KNN-LM (Khandelwal et al., 2020), and RAG (Lewis et al., 2020b). From the angle of the LLMs, Zheng et al. (2023) found that providing contexts to LLMs can help them generate truthful answers.

In this paper, we demonstrate the capability of NUGGET2D from two perspectives. In language modeling (section 5) and summarization (section 6.3), NUGGET2D is shown to generate a highly condensed representation of the context, while the results in autoencoding (section 4) and question answering (section 6.2) reflect that the details of the contexts can be recovered from nuggets. Moreover, in section 6.2 we show that NUGGET2D trained with text continuation preserves the capability of instruction following. This demonstrates LLMs can encapsulate more of their input into fewer hidden states than previously realized, suggesting a new direction for efficient foundation models.

## 8 CONCLUSION

Prior work introduced the NUGGET model, demonstrating that a compressed representation derived from an encoder-decoder framework still allowed for tasks like reconstruction of the input. Motivated by the community's shift to decoder-only LLMs, we adapt this framework to models such as LLAMA. While NUGGET utilizes states drawn from the last layer of an encoder as a compressed representation, our NUGGET2D solution employs a representation that preserves state from all layers of a model for a given token. We demonstrated that these "2 (D)imensional nuggets" support a variety of tasks, such as language modeling and text summarization, while offering significant savings and scalability to longer input contexts. Future work will explore more specialized versions of this proposal for optimizing results on individual applications, such as in dialog, supervised fine-tuning, reinforcement learning with human feedback, and in-context learning.

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

# A    OPTIMAL NUGGET SELECTION

The nugget selection module, i.e. `Scorer`, is learned through the residual connection introduced in section 2.3. With gradient signal from the self-attention, `Scorer` tends to select the tokens that are mostly attended by the decoder (parameterized by $\theta$), as we analyzed in eq. (6). However, it remains a question whether the selection is optimal. Here we provide an empirical estimate of the gap between the optimal nugget selection and `Scorer`.

Suppose we select $k$ nuggets out of $n$ tokens, we define a selection as a set of indices

$$\mathcal{I} = \{i_1, i_2, \ldots, i_k\}, \quad 1 \le i_j \le n.$$

From the definition we can see $\mathcal{I} \subseteq \{1, 2, 3, \ldots, n\}$. We further define the optimal selection $\mathcal{I}^*$ as the selection that achieves *the best performance* on a downstream task, e.g. lowest perplexity for language modeling. We denote the selection of `Scorer` as $\bar{\mathcal{I}}$. We want to answer two questions: How similar are $\mathcal{I}^*$ and $\bar{\mathcal{I}}$, and what is the performance gap between $\mathcal{I}^*$ and $\bar{\mathcal{I}}$?

Finding $\mathcal{I}^*$ is a non-trivial combinatorial optimization problem. The only possible solution, as we know, is to enumerate $\binom{n}{k}$ different nugget selections, which is infeasible for large $n$ and $k$. Therefore, we approximate $\mathcal{I}^*$ with a greedy algorithm. The basic idea is to start with $\mathcal{I} \leftarrow \bar{\mathcal{I}}$. Iteratively, for each index $i \in \mathcal{I}$, we replace it with an optimal index from the un-chosen indices so that it achieves the best downstream performance. We formalize it in algorithm 1 with an example downstream task of language modeling.

---

**Algorithm 1** A greedy algorithm to find the "optimal" selection $\mathcal{I}^*$.

---

**Input:** $k$ (number of nuggets) and $n$ (number of tokens) ($0 < k \le n$), encoder outputs $\mathbf{x}_{1:n}$
**Output:** A selection $\mathcal{I}$ and the corresponding LM perplexity $b$

   Initialize $\mathcal{I} = \{i_1, i_2, \ldots, i_k\}$ with `Scorer`.
   Perplexity $b \leftarrow \texttt{Decoder}(\mathbf{x}_{1:n}, \mathcal{I})$                   ▷ Lowest perplexity so far
   **for** $i \in \mathcal{I}$ **do**
      **for** $i' \in \{1, 2, \ldots, n\} \backslash \mathcal{I}$ **do**         ▷ All possible replacements from unchosen indices
         $\mathcal{I}' \leftarrow (\mathcal{I} \backslash \{i\}) \cup \{i'\}$                 ▷ Replace $i$ in $\mathcal{I}$ with $i'$
         Perplexity $b' \leftarrow \texttt{Decoder}(\mathbf{x}_{1:n}, \mathcal{I}')$
         **if** $b' < b$ **then**            ▷ If $i'$ is better than $i$, make the replacement permanent
            $b \leftarrow b', \mathcal{I} \leftarrow \mathcal{I}'$
         **end if**
      **end for**
   **end for**

---

We conduct experiments with the NUGGET2D checkpoint in section 5. We compress a sequence of up to 128 tokens into nuggets with a nugget ratio of 10%. We present the model with another 64 tokens without compression. The model is required to predict the next 64 tokens, and we measure the subword-level perplexity of NUGGET2D. Because algorithm 1 contains 2 for loops and is expensive to execute, we only sample 1000 documents from the test set of WikiText-103 (Merity et al., 2017).

To measure the difference between $\bar{\mathcal{I}}$ and $\mathcal{I}^*$, we count how many elements are replaced in $\bar{\mathcal{I}}$ with algorithm 1. On average, 24.7% nugget tokens are replaced, meaning `Scorer` is roughly 75.3% "correct". After replacing $\bar{\mathcal{I}}$ with $\mathcal{I}^*$, the overall subword-level perplexity is improved from 7.74 to 7.13, or $\mathcal{I}^*$ is roughly 7.9% better than $\bar{\mathcal{I}}$ in terms of downstream task performance.

In conclusion, we conduct experiments to show that `Scorer` is adequate to select nuggets as it can achieve similar performance as a decoder-aware optimal nugget selector.

# B    TRAINING SCORER WITH STRAIGHT-THROUGH ESTIMATOR

In section 2.3, we propose to build a residual connection between nugget scores and the self-attention of the decoder transformers to back-propagate gradients to `Scorer`. We copy it here:

$$\xi_{i,j}^l = \frac{1}{\sqrt{d}} \left[ \left( \mathbf{W}^{\mathrm{Q}} \mathbf{x}_i^l \right)^\top \left( \mathbf{W}^{\mathrm{K}} \mathbf{x}_j^l \right) + s_j \right], \quad i < j. \tag{15}$$

However, adding nugget scores to the attention logits affects the behavior of self-attention. In practice, we found that the nugget score (i.e. $s_j$ in eq. (15)) is of a similar scale of attention logits (i.e. $\left(\mathbf{W}^{Q}\mathbf{x}_i^l\right)^\top \left(\mathbf{W}^{K}\mathbf{x}_j^l\right)$ in eq. (15)) and the nugget scores are usually positive, meaning that nugget tokens draw more attention because of the addition of $s_j$.

Alternatively, we propose to remove the effect of $s_j$ in eq. (15) from the forward propagation of the neural network:

$$\xi_{i,j}^l = \frac{1}{\sqrt{d}}\left[\left(\mathbf{W}^{Q}\mathbf{x}_i^l\right)^\top \left(\mathbf{W}^{K}\mathbf{x}_j^l\right) + s_j - \texttt{StopGrad}(s_j)\right], \quad i < j, \tag{16}$$

where $\texttt{StopGrad}$ means detaching it from the computation graph. Therefore, $s_j$ receives the same gradients as the attention logits, but they do not impact the attention distribution. This is akin to the "straight-through estimator" discussed in Bengio et al. (2013).

In practice, the behavior of both eqs. (15) and (16) is similar: They quickly converge — the nugget selection is stable after 3000 steps of training — and they have similar performance on the autoencoding experiments in section 4. One can even remove the residual connection in eq. (15) when the $\texttt{Scorer}$ converges (after 3000 steps of training) and the model can quickly adapt to the new attention pattern. We speculate that Llama-7b permits more flexibility because of its tremendous size.

## C  A DETAILED COMPARISON BETWEEN NUGGET AND NUGGET2D

In this section, we show more details on the architecture of NUGGET and a succinct comparison between NUGGET and NUGGET2D. While Qin & Van Durme (2023) provides a thorough discussion of NUGGET.

**Applicability**  Qin & Van Durme (2023) can be applied to the encoder-side of an encoder-decoder transformer and NUGGET can be applied to decoder-only transformers, such as LLAMA.

**The form of nuggets**  The decoder exclusively attends to the last-layer representation of the encoder side, thus each nugget in Qin & Van Durme (2023) is the last-layer representation of its corresponding token ($\mathbf{z}_i \in \mathbb{R}^d$). For a decoder-only transformer, each future token attends to all-layer hidden states of past tokens. Therefore, a nugget in NUGGET2D is comprised of the hidden states of the corresponding token in all the layers ($\mathbf{z}_i \in \mathbb{R}^{L \times d}$, $L$ is the number of layers).

**The learning of nugget selection**  Both Qin & Van Durme (2023) and NUGGET2D have a nugget selection module $\texttt{Scorer}$. In Qin & Van Durme (2023), $\texttt{Scorer}$ takes features from the $l$-th layer of the transformer encoder, thus they freeze the first $l$ transformer layers to make the training stable. In NUGGET2D, $\texttt{Scorer}$ takes features from a frozen transformer decoder with $l$ layers that are initialized from LLAMA, thus all the layers of the encoder ($\phi$) and decoder ($\theta$) are trainable.

Although Qin & Van Durme (2023) uses a transformer encoder to represent text, a dual decoder must be equipped during the training phase to learn the nugget selection module. Suppose $\texttt{Scorer}$ is used to select nuggets, which outputs a score $s_i \in \mathbb{R}$ for the $i$-th token, it is plugged into the *cross-attention* to receive gradients:

$$a_{i,j} = \frac{1}{\sqrt{d}}\left[\left(\mathbf{W}^{Q}\mathbf{x}_j^{\text{tgt}}\right)^\top \left(\mathbf{W}^{K}\mathbf{z}_i\right) + s_i\right],$$

where $a_{i,j}$ is the attention score from the $j$-th token $\mathbf{x}_j^{\text{tgt}}$ on the target side to the $i$-th nugget $\mathbf{z}_i$. Note that the auxiliary decoder of Qin & Van Durme (2023) is discarded after training.

In contrast, NUGGET2D directly works on a transformer decoder, thus the training of the $\texttt{Scorer}$ in NUGGET2D is straightforward. In NUGGET2D, the residual connection between $\texttt{Scorer}$ and the *self-attention* is built to learn nuggets (refer to eq. (5). NUGGET2D does not require an auxiliary module to learn nuggets.

**Flexibility on text compression**  Qin & Van Durme (2023) compresses the entire text with a transformer encoder. NUGGET2D can compress any part of the text. Similar to our experiments on sections 5 and 6, one may flexibly choose to compress a certain part of the text.

Table 5: Parameter count of NUGGET2D . We do not distinguish `Llama-7b`, `Llama-2-7b`, and `Llama-2-7b-chat` here as they have the same architecture. The parameters of the encoder and decoder are counted as additional parameters with LoRA compared to the base model.

| Module | #parameters | Percentage | Trainable |
|---|---|---|---|
| Llama-7B | 6.74B | 99.01% | no |
| encoder (w/ LoRA, $\phi$) | 25.2M | 0.37% | yes |
| decoder (w/ LoRA, $\theta$) | 25.2M | 0.37% | yes |
| `Scorer` ($\varphi$) | 16.8M | 0.25% | yes |
| soft prompt ($\theta$) | 4,096 | <0.0001% | yes |

**Streaming decoding**   Due to the nature of the transformer encoder, Qin & Van Durme (2023) can only encode the text. Its decoder can learn to decode new texts, but it cannot re-encode them to nuggets in an efficient way. NUGGET2D , with parameter re-assignment described in section 2.5, can naturally decode new texts and re-encode them auto-regressively, achieving streaming decoding.

**Nugget type encoding**   Qin & Van Durme (2023) passes a "type embedding" during the encoding process to inform the model what tokens are selected as nuggets (eq 8 in their paper). Empirically we do not find any improvement after introducing the "type embedding", thus our experiments in the main paper are conducted without type embeddings. We speculate there are 2 reasons behind this:

1. An encoder transformer may not naturally concentrate context information into a few tokens, as the cross-attention from the decoder side always covers all token representations. In this case, a type embedding can guide the model to encode context semantics in nugget tokens. However, during the pretraining phase of a decoder-only transformer, the embedding of each token is used to predict the next token, thus it naturally gathers context information. In this case, a type embedding is redundant.

2. In the architecture of autoregressive NUGGET2D in section 2.5, nugget tokens are encoded with parameter $\phi$ instead of $\theta$ that are used for non-nugget tokens. A separate type embedding to distinguish nugget tokens is obviously unnecessary in this case.

## D   IMPLEMENTATION & TRAINING DETAILS

### D.1   IMPLEMENTATION

The training pipeline of NUGGET2D is implemented with the PyTorch (Paszke et al., 2019) and Pytorch Lightning package (Falcon & The PyTorch Lightning team, 2019). We use the ZeRO stage-2 provided by the DeepSpeed Rasley et al. (2020) package with mixed precision to accelerate the training. The implementation of NUGGET2D is based on the huggingface/transformers package (Vaswani et al., 2017). We used the implementation of huggingface/peft (Mangrulkar et al., 2022) for LoRA (Hu et al., 2022). Our dataset reader uses huggingface/datasets (Lhoest et al., 2021).

A soft prompt is involved for the training objectives of autoencoding and text continuation. The soft prompt is treated as part of the model parameters. Each soft prompt contains 1 soft token to be trained. We empirically found that the number of soft prompt tokens does not have a major impact on model performance.

### D.2   HYPERPARAMETERS AND TRAINING DEVICES

For all the experiments, we follow the training setup of Touvron et al. (2023b) and use an Adam optimizer (Kingma & Ba, 2015) with a learning rate of $1 \times 10^{-4}$, $\beta_1 = 0.9$, $\beta_2 = 0.95$, and $\epsilon = 10^{-5}$. We use a cosine learning rate scheduler (Loshchilov & Hutter, 2017) with warmup of 2k steps, and the period of the cosine annealing function is set as 150k steps. We train the models on 16 NVIDIA Tesla V100 GPUs (32 GiB), each with a batch size of 1. Gradients are accumulated for 2 batches before the execution of the optimizers. All the models are trained until early stopping because of the convergence of the loss on the validation set.

### D.3 NUMBER OF PARAMETERS

In this section, we enumerate the number of parameters in NUGGET2D , as shown in table 5. Except for the frozen Llama model, NUGGET2D has an encoder and decoder, which contains additional parameters to the Llama model with LoRA (Hu et al., 2022) (rank = 32), a scorer (2-layer feedforward neural networks), and a soft prompt that adds a special token to the embedding matrix.

For the experiments in section 5, we use LoRA to train COMPRESSIVE , which contains a decoder and a soft prompt as we shown in table 5. However, compared to the size of LLAMA, the trainable parameters of both NUGGET2D and COMPRESSIVE are significantly fewer ($<1\%$).

## E EXAMPLE TEXT FOR NUGGET SELECTION ANALYSIS

We sample a passage from Wikipedia and run `Scorer` on the text, where we set the nugget ratio $r$ as 0.1. The results are shown in fig. 7.

The Brooklyn Nets have built themselves up from next to nothing. Devoid of anything close to an asset before 2015, the Nets had to make something out of nothing. They have done so indeed, loading the roster and asset cupboards simultaneously. Unfortunately, just as quickly as Marks acquired youngsters, he must also decide which ones should stick around. It's an arduous exercise, and even tougher for a team far from contention. Most teams reach this stage just as they are close to playoff-caliber. The Nets do not have this luxury, and must evaluate with a much longer view than the average young team. Put simply, they must think like a contender before becoming one. Luckily, the current roster has distinct tiers of young players in terms of their long-term potential. Eight of the nine under-25 players can be split into two tiers. Locks The group of definite keepers is relatively simple. These players have the most potential of the current Nets. Although D'Angelo Russell has gone through some rough patches, he has displayed enough promising signs to warrant the "keeper" status. His crafty ball-handling, scoring off the dribble, shooting off the catch, and great passing vision all make him an ideal fit for Kenny Atkinson's attack. Being the No. 2 overall selection in a draft is typically enough credibility to keep a player around, but Russell has shown legitimate flashes of star potential as well. Giving up on him now would be a fatal mistake. Jarrett Allen, a rookie center from the University of Texas, has done a wonderful job in his specialized role. With superb athleticism that allows him to protect the rim and switch onto perimeter attackers, Allen is quite capable of captaining a modern defense. This athleticism helps him on offense as well, as he gets plenty of lobs to finish pick-and-roll plays. When in doubt, the guards can chuck it up to him for an easy deuce. The vertical dimension of basketball is rarely appreciated.

Figure 7: Example texts processed by the `Scorer` of NUGGET2D . Darker texts have a higher score than light texts. The tokens in green background are selected as nuggets.

## F PROMPTS USED IN THE PAPER

The prompt used by the LMSUMM method to generate a summary for a given text is:

```
[INST]
Please summarize the following text into $WORD words:
$TEXT
[/INST]
```

We replace `$WORD` with $\lceil n \cdot r \rceil$, where $n$ is the number of words (counted by spaces) and $r$ is a desired ratio (in section 6, $r$ is 0.1).

In the SQuAD experiment (section 6.2), a prompt is used to answer a question given a document:

```
[INST]
$DOCUMENT
Based on the provided document, answer the following question:
$QUESTION
[/INST]
```

We replace `$DOCUMENT` with the context document and `$QUESTION` with the question.

In the summarization experiment (section 6.3), we use the following prompt:

```
[INST]
$DOCUMENT
Please summarize the above document in one sentence.
[/INST]
```

We replace `$DOCUMENT` with the document to be summarized.

## G    NORMALIZATION ALGORITHM FOR SQUAD ANSWERS

The output of the language model tends to have tokens other than the answer or have different forms. For each pair of model output and SQuAD answer, we apply the following rules:

- Convert all English numbers to digits. E.g. convert "two" to "2".
- Replace all punctuation marks with spaces.
- Remove side spaces on both sides.
- Lowercase the string.

After these steps, a program is used to check if the model output contains the answer. We restrict the model to generate up to 64 tokens in case they generate many tokens to hit the answer. [11]

---

[11]They rarely do, as they are not optimized to cheat SQuAD.

