# OpenReview forum: "Nugget 2D: Dynamic Contextual Compression for Scaling Decoder-only Language Models"
_ICLR.cc/2024/Conference — Submitted to ICLR 2024_

### Official Review · Reviewer_TZ9E · 2023-11-02

**Soundness:** 3 good
**Presentation:** 3 good
**Contribution:** 2 fair
**Rating:** 6
**Confidence:** 3

**Summary:**

This paper extends the Nugget approach [Qin, Durme, '23] from encoder-decoder architectures to decoder only architectures and explores its applications to compression of context in auto-encoding, auto-regressive decoding and downstream tasks like QA and summarization.
In all applications, the context to be compressed is encoded by a Nugget2D encoder, which seems to be a vanilla transformer encoder (initialized by some Llama variant LLM). The encodings are passed through a scorer and embeddings from the top-k scores are selected (similar to the original Nugget paper). Finally a separate LM (again initialized using some Llama variant LLM) operates on the actual task, encodes prompt/question and decodes the answer. This LM is trained to attend to only the selected nugget encodings. To easily propagate gradients to the scorer, during this attention mechanism between LM and nugget encoder, the scores are added to the attention logits (similar to the original nugget paper). The LM, nugget encoder and scorer are finetuned using PEFT.
The paper demonstrates that on the task of auto-encoding, Nugget2D outperforms ICAE, which allocates fixed size memory for compression, on the auto-regressive LM task, Nugget2D outperforms "Compressive" baseline, which pools the embeddings of context divided into constant sized chunks. On the QA task, their model retains 90% of the performance of an uncompressed baseline at 5x compression. Interestingly on the summarization task, their model outperforms baseline without an compression.

**Strengths:**

Main strength of the paper is to show that the original Nugget approach can be scaled and adapted to be used by LLMs. The technique is also adapted to various use cases: auto-regressive LM, auto-encoding and downstream tasks. They show strong results compared to baseline approaches considered in the paper.

**Weaknesses:**

The weaknesses of the paper are as follows:
1. The novelty of the key idea. Extending nugget to nugget2D seems to be a straightforward extension by changing the residual connections.
2. Some experimental results/comparisons are not clear.
    a) The results of the section 4.2 are not provided in a table, but depicted in a figure: figure 4. I found Figure 4 to be hard to    interpret and not clear at all. What do the bars represent there, ICAE bar is missing for x axis values 100, 200, 300?
    b) For results in 5.2, are the number of trainable parameters for Compressive and nugget 2D comparable? nugget2d employs an encoder and an LM, whose parameters are fine-tuned separately. How does this compare to compressive?
3. Some missing details about the method and training:
    a) Is "informed nugget encoding" used in the original nugget work still applied in this work?
    b) In section 2.5, Choice of k, its mentioned that the scorer is not trained on the autoregressive text continuation task, but taken from     the autoencoding experiments, why is that?

**Questions:**

1. In Fig 2, the attention pattern of the LM is shown, what is the attention pattern within Nugget2D?
2. In equation 4, do (i, j) represent all possible indices, or are some indices restricted to a subset chosen by the scorer?
3. Is "informed nugget encoding" used in the original nugget work still applied in this work?
4. In section 2.5, Choice of k, its mentioned that the scorer is not trained on the autoregressive text continuation task, but taken from     the autoencoding experiments, why is that?
5. For results in 5.2, are the number of trainable parameters for Compressive and nugget 2D comparable? nugget2d employs an encoder and an LM, whose parameters are fine-tuned separately. How does this compare to compressive?

---

> ### Author Response · Authors · 2023-11-15
> **Replies to your comments**
>
> Thank you for a detailed summary of the paper and for acknowledging the contribution of our paper! We made revisions to our paper according to your comments. Due to the space limit, we put some of the new contents in the appendix. Below are our answers to your questions:
>
> ## Novelty of the key idea
>
> We argue that Nugget2D is a non-trivial extension of the Nugget, not only changing the form of nuggets and applying to decoder-only transformers:
>
> - It has a redesigned Scorer module, which does not need to freeze transformer encoder parameters as Nugget does. Also, to train the scorer, Nugget requires a dual decoder, which will be discarded. In contrast, Nugget2D builds residual connections between decoder tokens directly and does not need an auxiliary module.
> - Nugget2D is much more flexible than Nugget: It can flexibly compress any part of a text, achieving different levels of granularity. E.g., in section 6, we compress the document into nuggets but keep the prompt as tokens. Moreover, Nugget2D can do streaming language modeling, autoregressively compressing generated tokens.
>
> We provide a detailed comparison between the 2 models in Appendix C, covering technical details and their applications.
>
> ## Interpretation of autoencoding results
>
> We did not provide a table to present the result because the baseline ICAE uses boxplots, and we are unable to access their raw results, as we said in our footnote.  The bars in fig4 are the results of ICAE for sequence lengths of 100, 200, and 300. They are nearly 100%, so they are at the top of the graph. We added comments to the caption to clarify the plots.
>
> ## Fairness of comparison to Compressive Transformers
>
> We admit that Nugget2D contains more trainable parameters than Compressive Transformers. However, it is hard for us to change the number of parameters of Compressive Transformers without changing its design. On the other hand, the parameters of both models are much fewer than those of the LLaMA because of LoRA (fewer than 1%). Therefore, the difference between their sizes is nearly negligible. We provide a breakdown of the number of parameters in Appendix D.3, which may alleviate your concerns.
>
> ## Does Nugget2D use informed nugget encoding?
>
> No. We empirically found it is not helpful for downstream tasks. We have 2 speculations (a more thorough discussion can be found in Appendix C):
>
> - A decoder-only transformer uses the hidden state of the last token to predict the next token, thus it naturally gathers context information. Informing the model about the nugget selection might be unnecessary.
> - In the scenario of autoregressive Nugget2D, nugget tokens are encoded with separate parameters, thus a type embedding is obviously redundant.
>
> ## Scorer is not trained on the autoregressive text continuation task
>
> We did not train the scorer for autoregressive LM because there is a chicken-and-egg situation. Instead of taking a certain ratio of tokens as nuggets, autoregressive LM uses a threshold $\overline{s}$ on scores to select nuggets, which is decided by running scorer on all documents such that $\overline{s}$ selects out a certain ratio of tokens as nuggets *on average*. However, training the scorer needs a specified $\overline{s}$ so that the training pipeline can be started. Given the features fed into the scorer do not depend on $\theta$ or $\phi$, we find that re-using the scorer in section 2.4 is convenient and works well. We have updated the description in section 2.5 to make this point clearer.
>
> ## Attention pattern within Nugget2D
>
> The attention pattern within Nugget2D is a standard causal masking for decoder-only transformers. We have updated Fig 2 to make it clear.
>
> ## Indices in eq 4 (now eq 5)
>
> The $j$ indices in eq 5 are the indices of nuggets. The i indices in eq 5 are all tokens that attend to the nuggets. We updated the paper to make this point clearer.

---

> ### Author Response · Authors · 2023-11-21
> **Request for response**
>
> Dear reviewer TZ9E,
>
> Thank you again for your valuable time and insightful comments.
>
> Given that the author-reviewer discussion period is coming to a close soon, would you please let us know if our responses have resolved their concerns and if there are any other questions we can address to help recommend our paper?
>
> Best regards!
>
> Authors

---

### Official Review · Reviewer_Hb1S · 2023-11-02

**Soundness:** 3 good
**Presentation:** 2 fair
**Contribution:** 3 good
**Rating:** 6
**Confidence:** 3

**Summary:**

Qin and Van Durme (2023) proposes NUGGET for encoder-decoder transformer models where the input context to the encoder is "compressed" and represented by selected tokens to be conditioned on during decoding. This paper proposes NUGGET2D that extends the idea to decoder-only transformers. The "nuggets" are not just the last layer representations from the encoders, but representations of selected tokens in a context on every layer to be conditioned on in decoding the answer. Promising results are shown on autoencoding, language modeling, summarization, and question answering.

**Strengths:**

- The method is a novel and useful technique to process long contexts.
- The experiments on multiple important tasks show promising results.

**Weaknesses:**

- Typo: Before Eq (8) I think you mean "\textit{recent} tokens $w_{\tau+1:t}$" -- $t$ instead of $i$.
- The presentation can be improved and multiple questions need to be clarified. Please see Questions. -- Willing to increase the score once the questions are clarified.

**Questions:**

- Please define the NUGGET2D function in Sec. 2.2 NUGGET2D, including what is $\phi$, instead of letting readers figure out themselves. Later you use "NUGGET2D" as a function, but in Sec. 2.2 you only use it as a method name.
- Perhaps you could consistently use $\mathbf{z}$ to denote hidden states from $\phi$. In Eq. (2) (3) (4), are $\mathbf{x}_i^l$ computed using $\phi$ but not $\theta$?
- How does Qin and Van Durme (2023) train the Scorer? You should also add the explanation to 2.1 Background.
- You mention in footnote 4 that Scorer quickly converges, and then $s_j$ can be removed from Eq. (5). How do you decide when to remove it? Do you first train all trainable parameters (based on the task) with $s_j$, and then just continue to still train all trainable parameters without $s_j$? Removing $s_j$ will cause all activations to change suddenly; not sure if it's a problem.
- In Sec. 2.5, can you clarify how you pick $\tau$?
- Can you provide analysis about how much space are used by the extra model parameters when you introduce NUGGET2D?  I notice that $\phi$ and $\theta$ are not tied.
- The nugget tokens are picked based on $\phi$ but not $\theta$. Is it possible that some tokens' contextualized representations according to $\theta$ has much information but is not selected based on $\phi$?
- First paragraph on page 5 is confusing, e.g., you said $\theta$ in eq (11) but there's no $\theta$ in eq (11). Please clarify what re-assignment means with more understandable notations.
- Is it true that the model only processes $w_r+w_d$ tokens and ignore the beginning tokens if the context is long? Would it be helpful if, like in many long-context processing papers, you still have some "gist" tokens to summarize the beginning tokens?

---

> ### Author Response · Authors · 2023-11-15
> **Replies to your questions (1/2)**
>
> Thank you for your careful reading and all the useful suggestions! We have updated the paper according to your comments, with some major revisions in section 2.5. Because the main paper is already full, we put some discussions and a new experiment in the appendices. We hope our replies below can address your concerns:
>
> ## The definition of the Nugget2D function
>
> $\mathtt{Nugget2D}$ takes a sequence of texts as inputs. It first encodes the text with the encoder ($\phi$), and then uses the scorer ($\varphi$) to select nuggets. The output of $\mathtt{Nugget2D}$ is the hidden states of the nugget tokens.  We added eq 4 to the main paper to define the $\mathtt{Nugget2D}$ function. Thanks for your suggestion!
>
> ## Is $\mathbf{x}_i^l$ computed with $\phi$ not $\theta$?
>
> The hidden states $\mathbf{x}_i^l$ in eq 2 and 3 are computed with $\phi$ as they are on the $\mathtt{Nugget2D}$ side. In eq 4 (now eq 5), the $\mathbf{z}_j^l$  is computed with $\phi$ because it is a nugget, but $\mathbf{x}_i^l$ is computed with $\theta$ because it is on the $\mathtt{LM}$ side. We added subscript $\phi$ to eq 2 and 4 to clarify their parameters.
>
> ## How do Qin and Van Durme (2023) train the Scorer?
>
> They build a residual connection between the encoder and decoder to back-propagate the gradients to the scorer. In the revised paper, we added a description of the residual connection in Qin and Van Durme (2023) in section 2.1. Moreover, we discussed its architecture and compared it with the scorer training in Nugget2D in Appendix C.
>
> ## Removing the residual connection
>
> We empirically found that the nugget selection converges after 3000 steps of training. At this stage, removing the residual connection does not cause much performance drop, as the model can quickly adapt to the new attention weights. We did not find any significant performance difference after dropping the residual connection, therefore we did not detach the residual connection for the experiments in the main paper.
>
> Your concern is valid: Removing the residual connection causes the activation to suddenly change and affect the forward propagation. In Appendix B, we discussed an alternative solution to train the Scorer based on the *straight-through estimator*. The basic idea is to subtract  $s_j$ from the attention weights to cancel its effect on the forward pass. However, it does not show any improvement in the experiments of autoencoding. We reckon that LLaMA is so large that it can flexibly adapt to any new attention patterns.
>
> ## How do we pick $\tau$? Do we ignore the beginning tokens? How about gist tokens?
>
> Suppose Nugget2D is decoding the $t$-th token, then $\tau$ is set as $t-\omega_r$, which means we compress texts that are more than $\omega_r$ tokens away.
>
> We indeed ignore tokens that are $\omega_r+\omega_d$ tokens away to make the inference time finite for arbitrarily long sequences. However, we argue that the information is not truncated at $\omega_r+\omega_d$: A token can access the information that is $L\cdot(\omega_r+\omega_d)$ tokens earlier, where $L$ is the number of transformer layers, because the information is propagated not only from left to right but also from bottom to top. For example, let $\omega = \omega_d + \omega_r$, then $x_\omega^{l=1}$ can access $x_0^{l=0}$, $x_{2\omega}^{l=2}$ can access $x_\omega^{l=1}$, …, and $x_{L\cdot\omega}^{l=L}$ can access $x_{(L-1)\cdot\omega}^{l=(L-1)}$. It is similar to the idea of TransfomerXL, which is a recurrence-based transformer variant.
>
> One could use other compression methods like gist tokens to incorporate more distant information. However, our ultimate goal is to process infinitely long sequences with finite resources, and methods like gist tokens assume a finite sequence length. Moreover, the function of gist tokens overlaps with Nugget2D. For these reasons, we did not introduce a separate compressing method like gist tokens.
> We have included the above discussion in the revised paper.
>
> ## Analysis of additional model space
>
> It is true that $\phi$ and $\theta$ are not tied. Because they are both initialized from the same LLaMA checkpoint, we implement $\phi$ and $\theta$ with LoRA, which means we only add a marginal number of parameters to the LLaMA model. With a LoRA rank of 32, the number of all trainable parameters is fewer than 1% of LLaMA, and LLaMA itself is kept frozen. We added a parameter breakdown in Appendix D.3.

---

> ### Author Response · Authors · 2023-11-15
> **Replies to your questions (2/2)**
>
> ## Does $\theta$ prefer different nuggets as $\varphi$ does?
>
> We argue that $\varphi$ tends to select tokens that $\theta$ prefers, because the parameters of Scorer, $\varphi$, are trained with a residual connection from the $\mathtt{Nugget2D}\_\phi$ side to the $\mathtt{LM}\_\theta$ side.
> In section 2.3, we conclude that this training objective will assign higher scores to the nuggets that are most attended to by the target tokens (i.e., tokens encoded with $\theta$).
>
> To fully answer this question, we conducted an experiment in Appendix A, where we designed a greedy algorithm to approximate the *optimal nugget selection* that achieves the best downstream performance. We run experiments on language modeling and compare the performance of optimal nugget selection to the nugget Scorer. In conclusion, we found that the nugget selection with Scorer is only 24% different from the optimal selection, and their performance gap is less than 8%, meaning the selection done by Scorer is very close to the oracle.
>
> ## Parameter re-assignment
>
> We have re-organized this section to make the narrative more readable. To answer your question: In the scenario of section 2.3, texts are split into 2 segments (e.g. document and prompt + response), where $\mathtt{Nugget2D}\_{\phi,\varphi}$ is used to compress one part of text while the $\mathtt{LM}_\theta$ encodes the other part. In the scenario of autoregressive Nugget2D in section 2.5, the text is not split but some tokens are selected nuggets. Therefore, $\phi$ is used to encode the nugget tokens, while $\theta$ is used to encode the non-nugget tokens. With this re-assignment, Nugget2D is able to decode and compress new tokens in a streaming way.
>
> ## Other minor questions
>
> We have adopted other minor suggestions you pointed out, including using $\mathbf{z}$ to consistently denote the hidden states selected by $\phi$, fixing the typos you pointed out, and other minor issues.

---

> ### Author Response · Authors · 2023-11-21
> **Request for response**
>
> Dear reviewer Hb1S,
>
> Thank you again for your valuable time and insightful comments.
>
> Given that the author-reviewer discussion period is coming to a close soon, would you please let us know if our responses have resolved their concerns and if there are any other questions we can address to help recommend our paper?
>
> Best regards!
>
> Authors

---

> > ### Comment · Reviewer_Hb1S · 2023-12-03
> >
> > Thank you for the response. I have raised my score and hope these additional information can be integrated into the paper.

---

### Official Review · Reviewer_WuCW · 2023-11-22

**Soundness:** 2 fair
**Presentation:** 4 excellent
**Contribution:** 2 fair
**Rating:** 5
**Confidence:** 4

**Summary:**

# Response to the authors rebuttal

Thanks for your response.

My biggest concern to this paper is that its main claim on scaling LLM to handle long contexts is not sufficiently supported by its experiments. The authors conducted their experiments on NLP tasks with less than 1k tokens, including WikiText (640+64), SQuAD (231) and CNN/DailyMain (878). In the response, the authors also provides an experimental result, i.e. "the maximum context length for encoding is ~1200 tokens", and admit a fact, i.e. "it is still tough to conduct experiments with contexts longer than 2000". Although the authors say that limited computational resources, i.e. 32G V100, prevents Nugget2D from scaling to longer context, it is still doubtful if Nugget2D could process context with 10k or even more tokens on an advanced 80G GPU, given the theoretic memory complexity of
$O(m^2+n^2+mn)$. The complexity in autoregressive transformer seems a techniqucal error given an n-lengthed input, by ingoring the compressing process in Nugget2D.

In conclusion, I think this paper (Nugget2D with maximum 1200 tokens) does not contribute to long-context sequence modeling community, with observation on rapid progress in this field, such as RMT (1M tokens, 2023/4), AutoCompressors (30k tokens, 2023/5) and ChatGPT (4k->32k tokens, 2022/11-2023/6).

References:

Scaling Transformer to 1M tokens and beyond with RMT. Bulatov et.al. 2023/4

Adapting Language Models to Compress Contexts. Chevalier et.al. 2023/5

GPT4 Models: https://platform.openai.com/docs/models/gpt-4-and-gpt-4-turbo


# below is the original review

The authors propose NUGGET2D, a NUGGET extension, to solve long context modeling problem for existing LLMs.
The main idea of NUGGET2D is to dissect the contextual tokens into local and global parts.
The local tokens are fully kept for attention and the global tokens are progressively filtered out by a threshold score.
The proposed method is tested on three tasks, i.e. language modeling on WikiText and Pile, question answering on SQuAD and summarization on CNN/Daily Mail.

**Strengths:**

1. The idea is well-motivated that only part of contextual token representations are informative.
2. The redicual connection modification to ensure differentiability is novel.
3. The empirical results look good.

**Weaknesses:**

1. The experiments are not sufficient to support the main contribution of scaling autoregressive LLMs to long contexts, comparing with recent long-context LLM studies (e.g. RMT, LongLLAMA). See details in Questions.
2. The idea is quite simple by splitting context tokens into local and global tokens, where the distinguishable part is compressing global tokens with a layer-wise NUGGET.

[1] Scaling Transformer to 1M tokens and beyond with RMT. Aydar et.al. 2023
[2] Focused Transformer: Contrastive Training for Context Scaling. Tworkowski1 et. al. 2023

**Questions:**

1. The baseline is relatively limited. It would be good to see other long-context LLMs (e.g. LongLLAMA[1]) as baselines.
2. I wonder how the memory grows as the context length increases against other context scaling methods. It seems the memory complexity in training stage is $O(n^2 * r)$. What is the limit of the context length of your model?
3. The context length of downstream tasks is really short. Do you ever try other dataset for long-context downstream task evaluation, such as Multi-News[2], Narrative QA[3] and CUAD[4].
4. What does "100k tokens" on $5.2 stand for? Does it mean context length? How is the random selection process performed?


[1] Focused Transformer: Contrastive Training for Context Scaling. Tworkowski1 et. al. 2023

[2] Multinews: A large-scale multi-document summarization dataset and abstractive hierarchical model. Fabbri et.al. 2019

[3] The narrativeqa reading comprehension challenge. Kocisky et al. 2017

[4] CUAD: An expert-annotated nlp dataset for legal contract review. Hendrycks et al., 2021b

---

> ### Author Response · Authors · 2023-11-30
> **Reply to your comments**
>
> We thank the reviewer WuCW for your recognition of our contributions. Below are our responses to your questions:
>
> ## Compare to other baselines
>
> Thanks for pointing out the recently published works. However, there are technical difficulties for us to conduct experiments to establish new baselines. The paper LongLLaMA released their checkpoints based on OpenLLaMA 3B and 7B, while our experiments are based on LLaMA-2 7B. We are unable to expediently train another LongLLaMA on LLaMA-2 7B during the rebuttal period. Moreover, the LongLLaMA paper was released on July 6, 2023, 3 months within the submission deadline of ICLR 2024, thus we reckon Nugget2D is a contemporary work to LongLLaMA.
>
> For similar reasons, the experiments with RMT cannot be shown now. However, we want to emphasize that the idea of Nugget2D is orthogonal to these works, as integrating kNN or recurrent Transformer to Nugget2D is straightforward. We will bring discussion of their relationship in the next version of the Nugget2D paper.
>
> ## Novelty of the idea
>
> We argue that the novelty of Nugget2D is not simply splitting contexts and compressing the distant ones. Instead, the model can learn how to compress the distant tokens by selecting certain hidden states as representations of the past texts, while this discrete step can be learned in an end-to-end manner with a residual connection.
>
> ## Space complexity
>
> If Nugget2D is used as a context compressor, then the space complexity on the encoder side is still $O(n^2)$, and the complexity on the decoder side is $O(m^2 + m\cdot n)$, where $n$ ($m$) is the number of tokens in the encoder (decoder) side. Note that the nugget ratio $r$ is a constant throughout experiments so is only a coefficient.
>
> In the settings of autoregressive transformers, the space complexity of Nugget2D depends on the local window size $w\_r$ and distant windows size $w\_r$ and is $O((\omega\_r+\omega\_d)^2)$. If they are set as constant, the complexity is $O(1)$, though the actual computational overhead is still 7B-level.
>
> We conduct experiments with a single V100 GPU card with 32GiB memory, and the maximum context length for encoding is ~1200 tokens.
> We argue that Nugget2D is not designed to lower the theoretical space complexity but the coefficient $r$. Even with the same complexity, a compression ratio of 10x can still greatly increase the context lengths of LLMs in practice.
>
> ## Experiments on other datasets
>
> We thank the reviewer for pointing out several alternative datasets. We did consider running Nugget2D on datasets with longer sequence lengths, however, they are relatively small, and training LLaMA on them would cause significant overfitting. CNN DailyMail is the best trade-off we found, which contains 287k training examples with an average length of ~1000 tokens. Among the datasets you pointed out, the largest one is MultiNews, which only contains 44k examples, which we reckon is not sufficient for a 7B model.
>
> We also have technical difficulty running very long documents. We note that even with Nugget2D, it is still tough to conduct experiments with contexts longer than 2000 tokens due to our limited resources. We argue that Nugget2D can be viewed as a proof of concept. With sufficient resources and training on longer documents, it may achieve better performance on much longer documents.
>
> ## The sampling method for autoregressive LM experiments
>
> 100k indicates the total number of tokens for evaluation, summed over all evaluation documents. The context length is shown in Table 1.
>
> We held out a part of the Pile for evaluation. During the evaluation, we iterate over documents in the evaluation split, filtering out documents shorter than the required context lengths. We keep collecting documents until 100k tokens are used. We will clarify the sampling process in the paper and release the codes upon paper acceptance.

---

### Meta-Review · Area_Chair_AXfq · 2023-12-06

**Metareview:**

Summary and Contributions:
The submission introduces Nugget 2D, an extension of the Nugget approach aimed at improving the capability of large language models to handle long contexts in autoregressive tasks. The method applies a novel strategy to compress the context into two types of tokens—local and global—and selectively maintains information from the global tokens. The authors tested Nugget 2D across various tasks including language modeling, question answering, and summarization and demonstrated improvements over standard models, specifically on long-context scenarios.

Strengths:
- The reviewers acknowledge the challenging nature of the addressed problem (long-context modeling).
- The empirical results are promising, showing the capability of NUGGET2D to manage long contexts in some extend.
- The response to the initial reviews was comprehensive and added valuable clarifications to the work.

Weaknesses:
- The main concern of this paper lies in the evaluation part, namely authors should compare the proposed method with baselines in 1M setting. I think reviewer WuCW mentioned the LongLLAMA means you should test the proposed approach in a more challenge setting instead of a relatively normal setting to verify the effectiveness of this paper. I agree with Reviewer WuCW's comments that the main concern to this paper is that its main claim on scaling LLM to handle long contexts is not sufficiently supported by its experiments.

**Justification For Why Not Higher Score:**

See the main concern part.

**Justification For Why Not Lower Score:**

N/A

---

### Decision · Program_Chairs · 2024-01-16

Reject